# Artistoo, a library to build, share, and explore simulations of cells and tissues in the web browser

Inge MN Wortel[1,2†]*, Johannes Textor[1,2†]*

[1]Department of Tumor Immunology, Radboud Institute for Molecular Life Sciences, Nijmegen, Netherlands; [2]Institute for Computing and Information Sciences, Data Science, Radboud University, Nijmegen, Netherlands

**Abstract** The cellular Potts model (CPM) is a powerful in silico method for simulating biological processes at tissue scale. Their inherently graphical nature makes CPMs very accessible in theory, but in practice, they are mostly implemented in specialised frameworks users need to master before they can run simulations. We here present Artistoo (Artificial Tissue Toolbox), a JavaScript library for building 'explorable' CPM simulations where viewers can change parameters interactively, exploring their effects in real time. Simulations run directly in the web browser and do not require third-party software, plugins, or back-end servers. The JavaScript implementation imposes no major performance loss compared to frameworks written in C++; Artistoo remains sufficiently fast for interactive, real-time simulations. Artistoo provides an opportunity to unlock CPM models for a broader audience: interactive simulations can be shared via a URL in a *zero-install* setting. We discuss applications in CPM research, science dissemination, open science, and education.

**\*For correspondence:**
inge.wortel@ru.nl (IMNW);
johannes.textor@ru.nl (JT)

**Present address:** [†]Data Science, Institute for Computing and Information Sciences, Radboud University, Nijmegen, Netherlands

**Competing interests:** The authors declare that no competing interests exist.

## Introduction

A growing community of computational biologists uses simulation models to reason about complex processes in biological systems. The cellular Potts model (CPM, *Box 1*) is a well-established framework for simulating interacting cells. Originally proposed as a model for cell sorting (*Graner and Glazier, 1992*), the CPM has since been extended with a plethora of biological processes such as proliferation, apoptosis, cell motion, and chemotaxis – allowing CPM users to model diverse phenomena ranging from slime mould formation to blood vessel development, tumour growth, and cell migration (*Marée et al., 2007*; *Szabó and Merks, 2013*; *Hirashima et al., 2017*).

Nowadays, several mature modelling frameworks with CPM implementations exist, such as CompuCell3D (*Swat et al., 2012*), Morpheus (*Starruß et al., 2014*), Tissue Simulation Toolkit (*Daub and Merks, 2015*), and CHASTE (*Mirams et al., 2013*). Although CPMs are relatively efficient models, tissue-scale simulations still require substantial computational resources. For this reason, all of the abovementioned frameworks rely on the C++ programming language for computation steps, which requires them to be built for and installed on the user's native operating system.

Here, we present 'Artistoo' (Artificial Tissue Toolbox), a CPM framework built entirely in JavaScript. Although interpreted languages like JavaScript have classically been deemed too inefficient for running simulations, we found that this no longer holds: investments by major tech companies have tremendously improved JavaScript engines over the past years, to the point that our CPM now has no major performance disadvantage compared to existing C++ frameworks.

The JavaScript implementation of Artistoo opens up new possibilities for rapid and low-barrier sharing of CPM simulations with students, collaborators, and readers or reviewers of a paper. Unlike existing frameworks, Artistoo allows building simulations that run in the web browser without the

**eLife digest** Understanding complex systems, such as the weather or the spread of a pandemic, often relies on computational models that can simulate what is happening and what will happen next. Models like these can also be used to investigate biological processes. For example, cellular Potts models (or CPMs for short) are regularly used to simulate how cells move, self-organise to form tissues and respond to their surroundings.

Computational biologists use a range of specialist skills and software to create these models. However, this can make it difficult for people who do not have programming experience to interact with these simulations and incorporate them in to their own research. If more people could engage with these models, this could help foster closer collaborations and ultimately lead to better models of biological systems.

To make CPMs more accessible, Wortel and Textor created a toolbox called Artistoo that allows users to view and interact with simulations using just an internet browser. These simulations are very easy to interact with, and do not require any prior programming knowledge or specialised software. Viewers can input different parameters in to the simulation and watch in real time to see how this affects the biological system being modelled. Wortel and Textor showed that this toolbox can be used to build a range of different biological models and works just as fast as other, more complex programming tools.

Artistoo has many potential applications and is a valuable education, learning, and collaboration tool. It may also encourage more open science, as having more accessible computational models could help with peer review and make it easier to collaborate across different research fields. A similar approach could be used to provide access to many other types of models in biology and beyond.

need to install any software: Artistoo models run on any platform providing a standards-compliant web browser – be it a desktop computer, a tablet, or a mobile phone. These simulations can be published on any web server or saved locally and do not rely on any back-end servers being available. They can be made explorable, enabling viewers to interact with the simulation and see the effect of changing model parameters in real time.

In this paper, we will first briefly explain the key design principles behind Artistoo. We will then highlight applications in teaching, research, science dissemination, and open science where we envision that the zero-install, web-based architecture of our framework could be particularly useful.

## Results

### Implementation

Artistoo is a JavaScript library implemented as an ECMAScript 6 module, which can be loaded into an HTML page or accessed from within a Node.js command line application. Artistoo is an open-source library released under the MIT license and is freely available on GitHub at https://github.com/ingewortel/artistoo (copy archived at swh:1:rev:551ee76c3a972e86ac72e-d7e977356a75d091763; *Wortel and Textor, 2020*).

### Design philosophy

Computational modelling research involves two important, but distinct categories of researchers that tend to have different types of expertise. On the one hand, there are the model *builders*, the scientists designing the models and performing the research; these are typically computational biologists with at least some basic programming skills. On the other hand, there are the model *viewers*, members of the broader research community who should be able to access and understand these models once they are built; this group may also include biologists and students without programming expertise.

A major challenge in the design of modelling software is to cater to both these groups at the same time. Tools revolving around a front-end graphical user interface (GUI) are ideal for viewers (no programming required), but tend to lose some of the flexibility desired by builders (anything not yet

## Box 1. Cellular Potts models.

Cellular Potts models (CPMs) model cells and tissues as collections of pixels on a 2D or 3D grid, where each pixel has an 'identity' linking it to a specific cell or to the empty background.

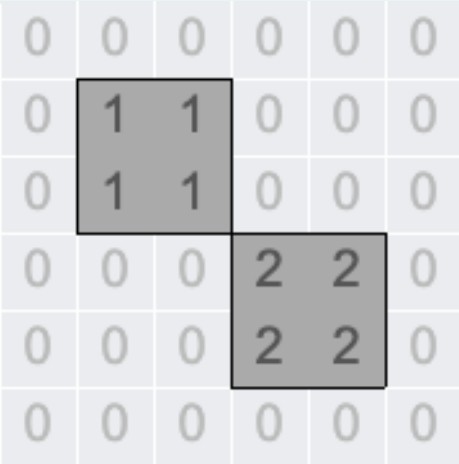

**Box 1—figure 1.** An example CPM grid with pixels belonging to the background (identity 0) or to one of two cells (identities 1 and 2).

Model dynamics arise from stochastic attempts to change these identities, for which the success rate $P_{\text{change}}$ is linked to the system's *global energy* or *Hamiltonian*, $H$. The energetic effect $\Delta H$ of the proposed change determines $P_{\text{change}}$: energetically favourable changes ($\Delta H<0$) always succeed, while the success rate of unfavourable changes ($\Delta H>0$) decays with the energetic 'cost': $P_{\text{change}} = e^{-\Delta H/T}$. Here, the 'temperature' $T$ is a model parameter controlling noise: a higher $T$ allows more energetically unfavourable changes to succeed.

CPM dynamics are thus controlled by the Hamiltonian $H$, an energy function defined by the modeller. $H$ can contain multiple terms to represent different biophysical processes, such as adhesion (interface energies) and shape elasticity (energetic penalties for cells stretching or compressing beyond a given size). One of the CPM's strengths is that almost any desired behaviour can be encoded into the model, provided that the modeller can come up with a suitable energy term. Furthermore, model energies can be linked to other equations (e.g. diffusion of some signalling molecule), allowing even more flexibility in the processes a CPM can simulate. For further details on typical energy functions and model dynamics, we refer the reader to Interactive Simulation 1 in Appendix 1.

As pixels can only have one cell identity at the same time, the property of *volume exclusion* emerges naturally in the model. This allows cells to interact with each other automatically. This – together with its flexibility and ability to capture detailed cell shapes – has made the CPM a popular tool for modelling cell–cell interactions and the resulting tissue dynamics (*Marée et al., 2007*; *Szabó and Merks, 2013*; *Hirashima et al., 2017*).

Nevertheless, like any model, CPMs have their limitations. For example, criticisms have included their lack of scalability, as well as difficulties in linking CPM parameters to measurable, real-world quantities. We note that ongoing developments in the field are addressing some of these concerns; for details on CPM strengths and limitations (and efforts to overcome these), we refer the reader elsewhere (*Tapia and D'Souza, 2011*; *Van Liedekerke et al., 2015*; *Magno et al., 2015*; *Rens and Edelstein-Keshet, 2019*; *Buttenschön and Edelstein-Keshet, 2020*).

implemented in the GUI typically becomes harder to do, and it becomes more difficult to automate simulations and post-processing). Vice versa, a more flexible coding-based tool is comfortable for builders but rapidly becomes inaccessible for most viewers.

The implementation in JavaScript allows Artistoo to resolve this problem by presenting each user group with a different interface (*Figure 1*). Model *viewers* access an HTML page provided by the model builder, which contains a model visualisation and interactive access to the most important parameters (improving transparency because viewers are not distracted by an overload of options they do not need). Such HTML pages are accessible in a zero-install setting, explorable via parameter sliders, and remain highly accessible: no knowledge of the Artistoo framework or model details is required for viewers to operate them.

Model *builders* create these web applications using the Artistoo framework. They can do this at different levels of complexity: Artistoo *users* build models via simple changes to configuration objects (requiring very little knowledge of Artistoo or programming), or by incorporating the many available methods in a few simple lines of code; this requires no in-depth knowledge of the framework 'under the hood' architecture while still providing high flexibility. Finally, Artistoo *developers* have the ultimate freedom to add custom plugins to the existing framework where needed. Only this group requires in-depth knowledge of the framework and slightly more advanced JavaScript skills. The online documentation at https://artistoo.net/ helps both these groups to get started with the framework.

## Approachability

The methods currently implemented in the framework allow users to simulate, visualise, and analyse a wide range of CPM models (*Figure 2A*). Our Github repository contains example code for models of various biological processes (e.g. simulations of tissues, cell migration, and cell interactions). First-time users can download these HTML pages and modify parameters without needing to learn the implementation details of the framework, or to have programmed in JavaScript before. Alternatively, the Simulation class provides default methods for setting up and visualising simulations, allowing users to get started with the library without having to set up this 'boilerplate' code themselves. Advanced users can instead build simulations from scratch and customise them using the many available options and methods. Once they become accustomed with the framework, they can also develop and plug in their own code modules (see 'Modularity and flexibility' below). An example interactive HTML simulation (*Figure 2B*) is included in Appendix 1, Interactive Simulation 2. Full documentation and a user manual with step-by-step tutorials are available at https://artistoo.net/.

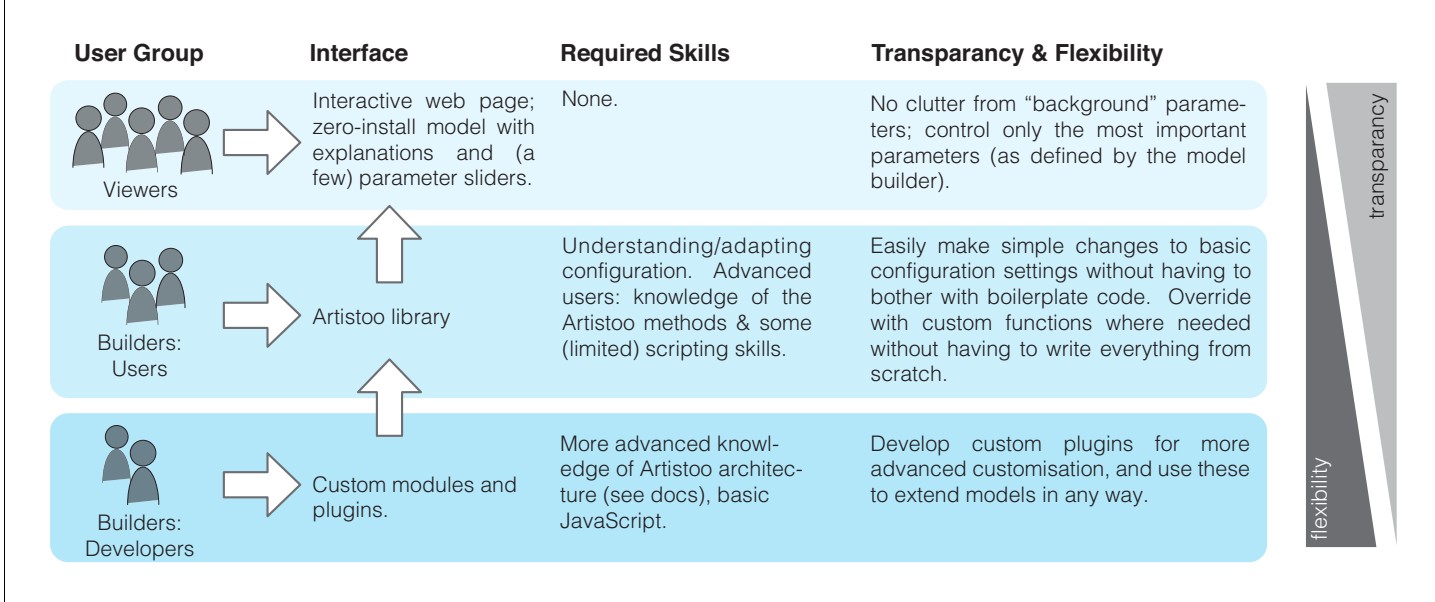

**Figure 1.** Artistoo provides different levels of access depending on the audience.

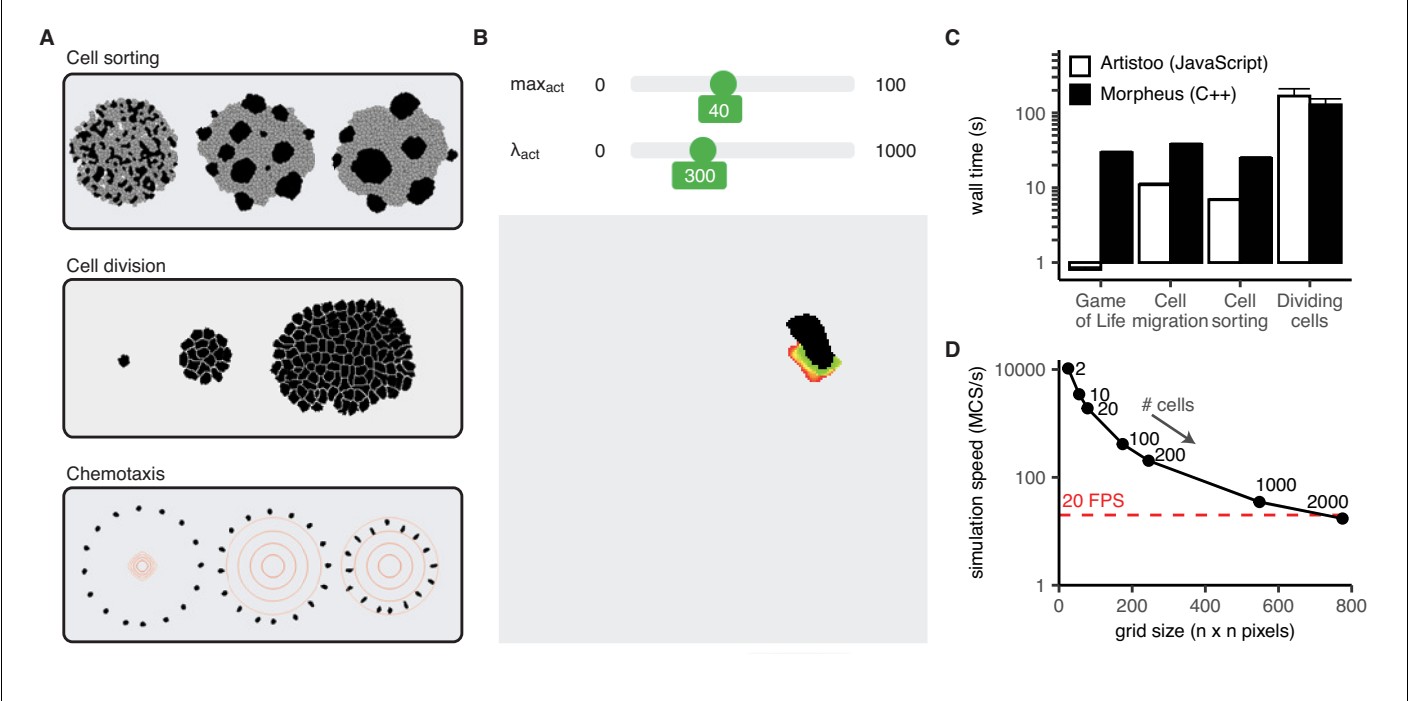

**Figure 2.** Artistoo supports interactive, 'real time' simulations of diverse biological processes. (**A**) Artistoo supports simulation of diverse biological processes; (**B**) users can interact with browser-based simulations via sliders, in real time. (**C**) Artistoo performance is comparable to that of the Morpheus framework. Data show wall times (mean ± SD of five runs) for four CPM models implemented in both frameworks (see Materials and methods for implementation details). (**D**) Scalability of the cell sorting simulation; simulation speed in Monte Carlo Steps per second (MCS/s) for different grid sizes (mean ± SD of five runs). Red line indicates 20 frames per second, a minimum speed required for a 'real-time' simulation for the human visual system. See also Appendix 1 for interactive versions of the simulations shown in (**B**)–(**D**). Artistoo Node.js scripts for the simulations in (**C**) and (**D**) are available on GitHub (*Wortel and Textor, 2020*).

The online version of this article includes the following figure supplement(s) for figure 2:

**Figure supplement 1.** Alternative representation of *Figure 2D* showing scaling with domain size.

## Modularity and flexibility

A typical CPM simulation consists of different types of components: the grid on which cells are simulated, the energy rules governing cell behaviour in the model, separate processes such as cell proliferation or diffusion, and the visualisation and quantification methods used to produce outputs. A key strength of the CPM is that it can be easily extended with custom terms to model specific processes. To facilitate such customisation, we have set up the code in a highly modular fashion. These modules can be combined freely to build a custom simulation. In addition, developers can supply their own custom modules – containing any of the aforementioned simulation components – to integrate with the framework and to share with other users.

## Performance and scalability

Although maximal performance is not a design goal of our framework per se, Artistoo should not be much slower than comparable frameworks either: running explorable simulations in real-time is only feasible if computations are reasonably efficient. Indeed, we implemented various simulations both in Artistoo and in Morpheus and found that both frameworks had similar performance (*Figure 2C*). In fact, Artistoo was slightly faster in all but one of these examples, although differences tended to be small; even in the case where Artistoo was slower (cell division), the difference in performance was not so large that real-time browser simulations became infeasible. Simulation speed scales linearly with the total number of pixels on the grid and does decrease for very large systems, but real-time simulations remain feasible for a reasonable range of grid sizes (*Figure 2—figure supplement 1*, *Figure 2D*). This would allow sharing of at least a reasonable prototype of larger-scale models.

## Portability

To make Artistoo more accessible for users familiar with other frameworks, we have built a prototype for an online tool that converts Morpheus model files into Artistoo code (https://artistoo.net/converter.html). In some cases, models may not (yet) be fully portable due to differences in the types of models supported; in that case, the tool returns the closest possible analogue and logs any changes it had to make, providing suggestions to help users further. This tool offers a starting point for any users who wish to build Artistoo web applications from their existing models. A similar tool converts Artistoo models to Morpheus XML, allowing users to continue in another framework (e.g. for upscaling models, multiscale models, etc).

## Applications

We here highlight a number of settings where Artistoo might complement other available modelling frameworks, focusing on the unique feature of Artistoo: it allows users to build and share explorable simulations in a *zero-install* setting. We discuss how this opens up novel opportunities of sharing CPM-based research and provide examples from our own work.

### Teaching

When organising practical computer work in the context of classroom teaching, getting software to work on every student's computer can consume a substantial amount of time and effort. Especially when teaching large classes in limited time, installing an entirely new modelling framework for a single-course assignment may not be appropriate. The *zero-install* feature of Artistoo might therefore be attractive for use of CPM modelling in the classroom. We frequently use the framework in teaching and found it feasible to let students run and understand CPM models in a workshop of just a few hours – even when students had no programming experience and were given just a single lecture on the CPM in advance. We provide an introductory assignment on the CPM in Application 1 in Appendix 2, which readers may use and adapt freely for their own courses, and refer to Interactive Simulation 1 in Appendix 1 for an interactive tutorial on the CPM.

### Communication and open science

While the move towards open science has prompted many to share their code with publications, understanding and using this code often remains challenging for readers who do not use similar models themselves. We envision that by sharing interactive Artistoo simulations via a simple URL, computational biologists can make their modelling research more accessible for the readers and reviewers of their papers; if readers can interact with model parameters themselves without the barrier of having to install special software, this may greatly improve the transparency of CPM research. This would allow others to evaluate these models more critically, as well as foster the exchange of ideas between scientists from different disciplines.

In addition, interactive simulations can help communicate CPM-based science at conferences or in classrooms. We frequently use the framework reveal.js (*Hattab and Contributors, 2020*) to build slideshows in HTML, in which live, interactive Artistoo demonstrations help explain how models work. Similarly, interactive simulations can be shared on a conference poster via a QR code, which other attendees can explore on their mobile phone. We provide examples of both in the Supplementary Materials (Application 2, Application 3 in Appendix 2).

### Research and collaboration

Although the CPM is extremely flexible in the types of behaviours it can model, it can be difficult to find the parameter ranges where these behaviours occur. We found that an interactive web page with instantaneous feedback, where the effect of changing parameters is visible in real time (*Figure 2* and Interactive Simulation 7 in Appendix 1), can substantially speed up parameter selection. This visual approach also picks up on unpredicted behaviours and artefacts (e.g. cell breaking) that are difficult to detect from numerical outputs alone. Moreover, we note that sharing these interactive pages allows us to tune parameters in collaboration with experimental biologists, helping us improve our models at an early stage. Thus, building a web-based prototype of a simulation can speed up parameter tuning and help obtain higher quality models.

Once a web-based prototype has been built and tuned, it can easily be ported to the Node.js JavaScript interpreter. This allows users to run a simulation as a command line application and store any desired simulation output locally. The resulting images or statistics can then be opened in other programs for further postprocessing, as demonstrated by two examples (Application 4 in Appendix 2).

## Discussion

The recent rise of the open science movement has changed the way research outputs are being shared and communicated. This may be especially important for computational models, which have classically been difficult to share because of the required software and coding skills. Transparent model sharing calls for new strategies to make models accessible for broader audiences.

Indeed, several such efforts have been made in recent years. The CPM framework CompuCell3D now hosts an online version on NanoHub, which users can access without installing software locally (*Gianlupi and Sego, 2021*). Beyond the CPM field, modelling frameworks like Tellurium (*Choi et al., 2018*) and PhysiCell (*Ghaffarizadeh et al., 2018*) have also created online access through NanoHub; these frameworks have also shown how such online models can be made interactive by using (variations of) Jupyter notebooks (see, e.g., *Macklin and Heiland, 2020*; *Medley et al., 2018*; *Somogyi, 2019*). The potential of explorable online web pages in communication and teaching is also demonstrated by the emerging practice to share R models in the form of Shiny apps (*Schönbrodt, 2014*; *Zehetleitner and Schönbrodt, 2015*; *Granjon, 2019*). And finally, the online collection of 'complexity explorables' (*Brockmann, 2020*) is a fantastic example of how to combine interactive online simulations with explanatory text to communicate modelling research.

With Artistoo, we now hope to open up this powerful avenue of model sharing for CPM research, allowing users to build online web pages and 'explorables' that combine interactive simulations with model explanations. We here show that the framework's performance (similar to that of existing frameworks in C++) is sufficient to allow for interactive CPM simulations. We have been developing the library for more than 5 years, also using it for robust simulation work in our research; see, for example, *Wortel et al., 2020*. We are continuing to develop the library for our own work and welcome suggestions and code contributions from the community.

We do not envision Artistoo to *replace* existing modelling software; rather, it can complement software directed at computational biologists and developers by letting users build explorable and sharable versions of a simulation. To facilitate this process, we have built a (prototype) tool to help users convert models between different frameworks (currently: Artistoo and Morpheus). Although Artistoo already offers a wide range of methods, it does not (yet) support all features of existing frameworks (Morpheus, CHASTE, CompuCell3D, Tissue Simulation Toolkit), such as solvers for reaction–diffusion equations or SBML-encoded intracellular signalling, or writing output in formats like VTK and HDF5. Nevertheless, Artistoo simulations are highly customisable, and a wide range of CPM models can already be constructed using the framework in its current state. The software's modular structure also makes it easy for future developers to extend it with custom code.

In summary, to the best of our knowledge, Artistoo is the first CPM simulation framework supporting interactive simulations in the web browser that can be shared via a simple URL, without requiring installed software or back-end servers. We hope that this will unlock avenues of sharing and communicating (CPM) simulations to much larger audiences.

## Materials and methods

**Key resources table**

| Reagent type (species) or resource | Designation | Source or reference | Identifiers | Additional information |
|---|---|---|---|---|
| Software, algorithm | Morpheus | Publication *Starruß et al., 2014* | RRID:SCR_014975 | version 2.1.0 |
| Software, algorithm | Artistoo | This paper, see also https://artistoo.net | RRID:SCR_020983 | version 1.0.0 |

This section contains implementation details of the simulations used to assess Artistoo (v1.0.0) performance. All simulations were run in the console mode (using Node.js, which contains the same JavaScript engine as the Chrome web browser). All Artistoo code is available in the repository https://github.com/ingewortel/artistoo-supplements/, as are interactive HTML versions of each simulation. Please visit https://ingewortel.github.io/artistoo-supplements/ for a web interface to access interactive simulations online. We refer to the provided code for details of the implementation, but summarise the most important settings here.

## Framework comparisons

To compare performance of Artistoo versus that of Morpheus (v2.1.0), we performed four different simulations in both frameworks. For this, we used the default examples provided with Morpheus and rebuilt similar simulations in Artistoo.

### Game of life

This is an implementation of the Game of Life, a Cellular Automaton (CA) of John Conway (see also Interactive Simulation 3 in Appendix 1). The simulation was run on a 50 × 50 pixel grid with random initial conditions. The simulation was run for 500 steps, storing a PNG image every 20 steps. This simulation is the Morpheus example *Miscellaneous/GameOfLife.xml* (version 4).

### Protrusion model

This model of a migrating cell implements an actin-inspired migration model (*Niculescu et al., 2015*) (see also Interactive Simulation 4 in Appendix 1). A single cell was seeded in the middle of a 200 × 200 pixel grid. Two obstacles of radius 10 were placed at a distance of 50 pixels to the left and right of the cell, respectively. Simulations were run for 15,000 MCS, logging the cell's centroid every 10 MCS and saving a PNG every 250 MCS. This simulation is the Morpheus example *CPM/Protrusion_2D.xml* (version 4).

### Cell sorting

This simulation implements the classical CPM model published by *Graner and Glazier, 1992* (see also Interactive Simulation 5 in Appendix 1). Fifty cells each of two cell types were seeded on a 200 × 200 pixel grid within a circle of radius 67 from the grid midpoint. Simulations were run for 2000 MCS, logging statistics every 10 MCS and saving a PNG every 100 MCS. This simulation is the Morpheus example *CPM/CellSorting_2D.xml* (version 4), where the *StopTime* field was changed from 2.5e4 to 2000.

### Cell division

A CPM linked to cell division (see also Interactive Simulation 6 in Appendix 1) was simulated on a 500 × 500 pixel grid. The grid was initialised with 20 cells in a circle of radius 35 surrounding the grid midpoint. Simulations were run for 40,000 MCS, logging the number of cells every 100 MCS and saving a PNG every 1000 MCS. This simulation is the Morpheus example *CPM/Proliferation_2D.xml* (version 4).

### Scalability of cell sorting

For the scalability simulations, simulations were run without outputting images. This allowed us to investigate the simulation speed separately from the time it takes to draw the entire grid. Note that if the drawing step becomes a limiting factor for running the simulation, it is always possible to speed up the process by drawing only once every few steps, or by choosing a more efficient drawing method (e.g. drawing only cell borders rather than entire cells).

Simulations contained 1, 5, 10, 50, 100, 500, or 1000 cells per cell type. The grid dimensions were adaptively scaled such that $x = y = \sqrt{1.5P_{\text{tot}}}$, with $P_{\text{tot}}$ the total number of pixels of all the cells. Cells were seeded within a radius $0.8\sqrt{P_{\text{tot}}/\pi}$ from the grid midpoint. Other settings were the same as in the cell sorting simulation described under Framework comparisons.

## Acknowledgements

The authors thank Nino van Halem and Ankur Ankan for their contributions to the code, Peter Linders for valuable feedback on an early version of the Artistoo manual, and Franka Buytenhuijs for thoroughly checking the interactive explorables. Funding This work was supported by KWF Kankerbestrijding (10620 to JT), a Vidi grant from NWO (192.084 to JT), and a PhD grant by the Radboudumc (to IW).

## Additional information

### Funding

| Funder | Grant reference number | Author |
| --- | --- | --- |
| KWF Kankerbestrijding | Young Investigator Grant (10620) | Johannes Textor |
| Nederlandse Organisatie voor Wetenschappelijk Onderzoek | Vidi Grant (VI.Vidi.192.084) | Johannes Textor |
| Radboud Universitair Medisch Centrum | Master-PhD grant | Inge MN Wortel |

The funders had no role in study design, data collection and interpretation, or the decision to submit the work for publication.

### Author contributions

Inge MN Wortel, Conceptualization, Data curation, Software, Formal analysis, Funding acquisition, Validation, Investigation, Visualization, Methodology, Writing - original draft, Project administration; Johannes Textor, Conceptualization, Resources, Data curation, Software, Formal analysis, Supervision, Funding acquisition, Validation, Investigation, Visualization, Methodology, Project administration, Writing - review and editing

### Author ORCIDs

Inge MN Wortel  https://orcid.org/0000-0003-3362-5229
Johannes Textor  https://orcid.org/0000-0002-0459-9458

### Decision letter and Author response

Decision letter https://doi.org/10.7554/eLife.61288.sa1
Author response https://doi.org/10.7554/eLife.61288.sa2

## Additional files

### Supplementary files

• Transparent reporting form

### Data availability

Source scripts have been provided for Figure 2.

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

## Appendix 1

### Interactive simulations

To showcase how interactive simulations can be used in teaching and communication, we have prepared a number of 'interactive simulations' and 'explorables'. These explain the CPM framework and the models used in *Figure 2*, and will also be made available on http://artistoo.net later.

All simulations described below are available on Github at https://github.com/ingewortel/artistoo-supplements (*Wortel and Textor, 2020*), which also contains instructions on how to use these files from your computer. For easy access, interactive simulations and applications are also accessible directly via https://ingewortel.github.io/artistoo-supplements/.

### Interactive simulation 1

This explorable explains the algorithm and dynamics behind the CPM for readers unfamiliar with this type of model.

### Interactive simulation 2

This explorable describes a CPM extension that lets cells migrate actively (*Niculescu et al., 2015*), as depicted in *Figure 2b*.

### Interactive simulation 3

This explorable describes a famous Cellular Automaton (CA) model called the 'Game of Life'; the classic model by John Conway that is also mentioned in *Figure 2c*.

### Interactive simulation 4

Interactive version of the 'Cell migration' simulation in *Figure 2c*. This model once again depicts a migrating cell as defined in *Niculescu et al., 2015* (see also Interactive Simulation two fur the full 'explorable' with an explanation of the model).

### Interactive simulation 5

Interactive version of the 'Cell sorting' simulation in *Figure 2c*. This classic model was the first CPM as developed by *Graner and Glazier, 1992*. Two populations of cells spontaneously sort themselves.

### Interactive simulation 6

Interactive version of the 'Dividing cells' simulation in *Figure 2c*; the explorable shows what happens when we couple division dynamics to the spatial information in a CPM.

### Interactive simulation 7

Interactive simulation of 'collective migration'. This simulation is an extension of Interactive Simulation 2, but now contains more than one cell and allows users to tune more different CPM parameters. See also Application 1 in Appendix 2. This simulation also serves as an example of how interactive CPM simulations can be used to tune parameters.

## Appendix 2

### Applications

All applications described below are available on Github at https://github.com/ingewortel/artistoo-supplements (*Wortel and Textor, 2020*). For easy access, interactive simulations and applications are also accessible directly via https://ingewortel.github.io/artistoo-supplements/.

### Application 1

An example exercise used to for teaching workshops on the CPM for beginning users. Readers are free to use this material in their own education. The file refers to an online simulation, but the same simulation is included as Interactive Simulation 7 in Appendix 1.

### Application 2

An example slideshow containing a live Artistoo simulation. The slides were built using the revealjs framework (*Hattab and Contributors, 2020*), which allows users to build slidesets in HTML.

### Application 3

A website accompanying a conference poster, which can be shared via a QR code on the poster itself. Please visit https://computational-immunology.org/inge/poster-cpmjs/ to view this example.

### Application 4

Two examples of how to use the Node.js version of Artistoo to export numeric outputs and images and to process these for downstream analysis using other programs. Both are available from our Github repository at https://github.com/ingewortel/artistoo-supplements (*Wortel and Textor, 2020*), see https://github.com/ingewortel/artistoo-supplements/tree/master/applications for usage instructions. The 'analysing-data/' folder contains a simulation of diffusion (so not a CPM); to show its numerical accuracy, the downstream analysis in R compares simulation output to an analytical solution. The 'making-movies/' folder contains a simulation of a migrating cell and shows how to make an animated movie of the produced images. Numerical outputs are also stored and used in a downstream R script to produce a cell track and a mean squared displacement curve.

