## [Decision Letter]

**Acceptance summary:**

This well written paper presents Artistoo, a software package allowing tissue simulations using the Cellular Potts framework, that fills an interesting niche: interactive, real-time simulations of complex multicellular systems that can run in a web browser, without any need for users to install or configure software. This enables new modes of education, science communication, research and multidisciplinary collaboration. This fully open-source software is impressive, and the supplied examples and tutorials are clean and beautifully fluid. The work should be of considerable interest to the *eLife* readership, particularly computational biologists and educators. The addition of markup language support both Morpheus to Artistoo and vice versa is fantastic. This will undoubtedly increase the adoption of Artistoo by the community. This is to my knowledge the first example of standardization across open-source CPM software packages.

**Decision letter after peer review:**

Thank you for submitting your article "Artistoo: build, share, and explore simulations of cells and tissues in the web browser" for consideration by *eLife*. Your article has been reviewed by 2 peer reviewers, one of whom is a member of our Board of Reviewing Editors, and the evaluation has been overseen by Aleksandra Walczak as the Senior Editor. The following individual involved in review of your submission has agreed to reveal their identity: Paul Macklin (Reviewer #2).

The reviewers have discussed the reviews with one another and the Reviewing Editor has drafted this decision to help you prepare a revised submission.

We would like to draw your attention to changes in our policy on revisions we have made in response to COVID-19 (https://elifesciences.org/articles/57162). Specifically, when editors judge that a submitted work as a whole belongs in *eLife* but that some conclusions require a modest amount of additional new data, as they do with your paper, we are asking that the manuscript be revised to either limit claims to those supported by data in hand, or to explicitly state that the relevant conclusions require additional supporting data.

Summary:

This is a well written paper that presents a software allowing tissue modelling using the Cellular Potts framework filling an interesting niche: interactive, real-time simulations of complex multicellular systems that can run in a web browser, without any need for users to install or configure software. As the authors describe, this enables new modes of education, science communication, and multidisciplinary collaboration. The software itself is impressive, and the supplied examples are clean and beautifully fluid. It is eye-opening that Javascript can run these models so well. The authors also did a fantastic and complete job in sharing their full source code, from the overall software down to individual scripts used to generate figures. The work should be of considerable interest to the *eLife* readership, particularly computational biologists and educators.

Essential revisions:

1. Suitability of the software for researchers:

a. Artistoo simulations do not appear to have any method to save data for external manipulation and archival. This makes their use somewhat less applicable to robust simulation-driven investigations, particularly where postprocessing and further analyses are required.

b. It is unclear if Artistoo-based models can be exported into other cellular Potts (CP) frameworks such as CC3D or Morpheus. This may leave researcher end users without a clear "upgrade path" after exploring model ideas in Artistoo and moving to larger simulations (e.g., larger or more complex domains), running simulations in high throughput on HPC resources, or adapting approximate Bayesian techniques for parameter estimation that require automating many simulation runs. Without an upgrade path, such users may wish to immediately begin in research-focused platforms rather than start with Artistoo and re-implement in another framework later.

c. Similarly, it is unclear if a model developed in Morpheus or CC3D can be directly imported into Artistoo. If such an import were possible rather than re-implementing models in Artistoo, research-focused users would be more likely to use Artistoo for scientific communication and outreach.

d. It would be fantastic if Artistoo would support the same markup language as Morpheus, allowing non-expert users to assemble complex models without writing a single line of code. If Artistoo would support the Morpheus ML, this would make all existing ``Morpheus models' also ``Artistoo models', meaning that Artistoo would become the standard for sharing CPM models with collaborators. Finally, adopting a common markup language between projects would be the first example of standardization across open-source CPM software packages.

2. Need for improved educational scaffolding:

The examples provided in the paper are excellent. However, they lack context on what the parameters mean or do. (For example, what are max_act_ and λ_act_ in the cell migration model?) This may limit the educational impact because users will be unclear on what to change, and how the parameters relate to cell biophysical processes.

The authors should include more background information with each model, define parameters, and give end users some idea of what to expect when parameters are changed. We have also found it useful to help guide a new user's exploration of a model by suggesting parameter sets and describing what they should see. This can serve as an educational scaffolding to help learners build and grow.

The authors' sample models should serve as a template to Artistoo users on best practices for communicating models to diverse audiences.

3. New developments in online cellular Potts simulators:

The authors should note that CompuCell3D has recently been ported to run interactively online in a web browser. See https://nanohub.org/resources/compucell3d. This recent development should be addressed in the paper.

4. Narrow review of interactive, "zero install" simulation frameworks:

The authors focus too narrowly by only comparing Artistoo with other cellular Potts frameworks, while the main use case for Artistoo is for interactively sharing and communicating complex simulation models online.

The authors should discuss non-CP frameworks that worked towards this, such as CC3D on nanoHUB (see above), online Tellurium (https://nanohub.org/resources/tellurium), current practice to share R models online as Shiny apps, and recent work to use xml2jupyter to automatically convert research-focused (command line) PhysiCell models to interactive Jupyter notebooks that can be shared as interactive webapps on nanoHUB (e.g., https://nanohub.org/tools/pc4cancerimmune). All of these serve similar purposes of creating zero-install, interactive versions of models for science education and communication. The authors should briefly discuss these to further contextualize their work.

5. While this is a more minor point, I would feel more comfortable if the supplementary information had convergence and accuracy testing. Are there limits on computational step sizes for numerically accurate simulations, particularly for large energies or when including diffusion processes?

---

## [Author Response]

Essential revisions:1. Suitability of the software for researchers:a. Artistoo simulations do not appear to have any method to save data for external manipulation and archival. This makes their use somewhat less applicable to robust simulation-driven investigations, particularly where postprocessing and further analyses are required.

We agree that the submitted manuscript did not explain well how Artistoo can be used by researchers. Artistoo simulations can not only be run in the web browser, but also in the command line using the Node.js JavaScript interpreter. Within Node.js, there are a huge amount of standard library functions and third-party packages (via the npm package manager) available that can be used to store data in a variety of different formats for further postprocessing. This is the recommended mode of operation for simulation-driven investigations, and we ourselves are regularly using Artistoo for complex simulation studies in our own work. To make this more clear, we have made the following changes:

– The manuscript now mentions this point explicitly in the section ”research and collaboration” (lines 203-207 on p6)

– Two examples have been added to the Supplementary Materials (Appendix 2, new Application S4) that show how to use Artistoo from Node.js to store data for further processing. In the Diffusion example, textual information is stored for further analysis in R. In the other example of a simulated migrating cell, PNG images are exported from Artistoo and further processed into a movie.

– We have added a citation to the manuscript (line 229 p7, Wortel et al. 2020) where we used Artistoo to perform thousands of simulations of migrating cells different 2D and 3D environments. Full source code for that work will soon be available at https://github.com/ingewortel/2020-ucsp.

b. It is unclear if Artistoo-based models can be exported into other cellular Potts (CP) frameworks such as CC3D or Morpheus. This may leave researcher end users without a clear "upgrade path" after exploring model ideas in Artistoo and moving to larger simulations (e.g., larger or more complex domains), running simulations in high throughput on HPC resources, or adapting approximate Bayeseian techniques for parameter estimation that require automating many simulation runs. Without an upgrade path, such users may wish to immediately begin in research-focused platforms rather than start with Artistoo and re-implement in another framework later.

It is indeed important that users are aware of the existing upgrade paths so they can make an informed choice before they begin working with our framework. To address this point, we made the following changes:

– As described in point 1a, the manuscript now places more emphasis on the fact that users can easily port simulations from the web browser to the Node.js platform; this allows users to automate simulations, perform many simulations in parallel, perform parameter screens, export data in various formats etc.

– We have developed a conversion tool that exports Artistoo models to the Morpheus XML syntax. While such a conversion is not straightforward given that there are some Artistoo features do not have Morpheus equivalents, our first prototype can already convert many simulations completely (e.g. several of the examples provided on artistoo.net/examples.html), and it tries to produce at least a reasonable approximation in many other cases. The tool will also log any changes made in the process and offers suggestions where additional manual tuning may be required. This tool is available on https://artistoo.net/converter.html, and is described in a new manuscript section “Portability” (lines 153-161 on p5)

c. Similarly, it is unclear if a model developed in Morpheus or CC3D can be directly imported into Artistoo. If such an import were possible rather than re-implementing models in Artistoo, research-focused users would be more likely to use Artistoo for scientific communication and outreach.

The reverse conversion from Morpheus XML to Artistoo is also covered by the new tool mentioned in point 1b; please see the response to 1b above. Generally, full conversion from Morpheus XML to Artistoo will only be feasible once our framework implements all features that are also supported by Morpheus, but our prototype can already convert some of the examples that are shipped with Morpheus and provides a good starting point for futher development.

d. It would be fantastic if Artistoo would support the same markup language as Morpheus, allowing non-expert users to assemble complex models without writing a single line of code. If Artistoo would support the Morpheus ML, this would make all existing ``Morpheus models' also ``Artistoo models', meaning that Artistoo would become the standard for sharing CPM models with collaborators. Finally, adopting a common markup language between projects would be the first example of standardization across open-source CPM software packages.

The online tool assists in translating Artistoo models into the Morpheus markup language and vice versa; please see the responses to 1b and 1c above.

2. Need for improved educational scaffolding:The examples provided in the paper are excellent. However, they lack context on what the parameters mean or do. (For example, what are max_act_ and λ_act_ in the cell migration model?) This may limit the educational impact because users will be unclear on what to change, and how the parameters relate to cell biophysical processes.The authors should include more background information with each model, define parameters, and give end users some idea of what to expect when parameters are changed. We have also found it useful to help guide a new user's exploration of a model by suggesting parameter sets and describing what they should see. This can serve as an educational scaffolding to help learners build and grow.The authors' sample models should serve as a template to Artistoo users on best practices for communicating models to diverse audiences.

We agree with the reviewers that in the examples included with the manuscript, we missed the opportunity to showcase the full potential of an interactive model. We have therefore added the suggested context to the interactive simulations provided in the Supplementary Materials (Appendix 1 and see https://ingewortel.github.io/artistoo-supplements/ ), transforming them from standalone simulations to full ”explorables”. These explorables, beyond showing how to use the technology, now also serve as an example of how to communicate models to diverse audiences. They will soon also be made available on https://artistoo.net and may be used freely as a teaching resource.

3. New developments in online cellular Potts simulators:The authors should note that CompuCell3D has recently been ported to run interactively online in a web browser. See https://nanohub.org/resources/compucell3d. This recent development should be addressed in the paper.

We thank the reviewers for bringing this to our attention and have now mentioned this in the manuscript discussion (lines 213-215 on p6).

4. Narrow review of interactive, "zero install" simulation frameworks:The authors focus too narrowly by only comparing Artistoo with other cellular Potts frameworks, while the main use case for Artistoo is for interactively sharing and communicating complex simulation models online.The authors should discuss non-CP frameworks that worked towards this, such as CC3D on nanoHUB (see above), online Tellurium (https://nanohub.org/resources/tellurium), current practice to share R models online as Shiny apps, and recent work to use xml2jupyter to automatically convert research-focused (command line) PhysiCell models to interactive Jupyter notebooks that can be shared as interactive webapps on nanoHUB (e.g., https://nanohub.org/tools/pc4cancerimmune). All of these serve similar purposes of creating zero-install, interactive versions of models for science education and communication. The authors should briefly discuss these to further contextualize their work.

The manuscript discussion now contains a more extensive review of sharing interactive models online (lines 213-223 on p6).

5. While this is a more minor point, I would feel more comfortable if the supplementary information had convergence and accuracy testing. Are there limits on computational step sizes for numerically accurate simulations, particularly for large energies or when including diffusion processes?

While numerical solution of continuous equation systems is not the main focus of Artistoo, it can be used to implement simple step-wise forward numerical solution schemes, if needed; it will then need to obey the same limits on step size and spatial resolution as other implementations of the same schemes. To illustrate this point, we have added a new example to the Supplementary Material (Appendix 2, Applicaton S4, accessible via https://github.com/ingewortel/artistoo-supplements/tree/master/applications, S4 > “analysing-data”); there, we use Artistoo to solve a diffusion equation that has an analytical solution, and verify that the solution obtained by our simulation closely matches the analytical solution as long as the standard Courant-Friedrichs-Lewy stability condition is fulfilled and the grid is not too coarse. This example simultaneously also illustrates how one can export data from Artistoo simulations for downstream statistical analysis (point 1a)